# *TSLP* and *TSLPR* Expression Levels in Peripheral Blood as Potential Biomarkers in Patients with Chronic Rhinosinusitis with Nasal Polyps

**DOI:** 10.3390/ijms26031227

**Published:** 2025-01-30

**Authors:** Emma Moreno-Jiménez, Natalia Morgado, Manuel Gómez-García, Catalina Sanz, María Gil-Melcón, María Isidoro-García, Ignacio Dávila, Asunción García-Sánchez

**Affiliations:** 1Instituto de Investigación Biomédica de Salamanca, 37007 Salamanca, Spain; emorenoj.ibsal@saludcastillayleon.es (E.M.-J.); nmorgado.ibsal@saludcastillayleon.es (N.M.); mgomezgarcia4.ibsal@saludcastillayleon.es (M.G.-G.); misidoro@saludcastillayleon.es (M.I.-G.); idg@usal.es (I.D.); chonela@usal.es (A.G.-S.); 2Microbiology and Genetics Department, Universidad de Salamanca, 37007 Salamanca, Spain; 3Clinical Biochemistry Department, Hospital Universitario de Salamanca, 37007 Salamanca, Spain; 4Instituto de Salud Carlos III, Red de Enfermedades Inflamatorias—RICORS, 28029 Madrid, Spain; 5Otorhinolaryngology and Head and Neck Surgery Department, Hospital Universitario de Salamanca, 37007 Salamanca, Spain; mgilmel@saludcastillayleon.es; 6Medicine Department, Universidad de Salamanca, 37007 Salamanca, Spain; 7Biomedical and Diagnostics Sciences Department, Universidad de Salamanca, 37007 Salamanca, Spain; 8Allergy Department, Hospital Universitario de Salamanca, 37007 Salamanca, Spain

**Keywords:** TSLP, TSLPR, N-ERD, CRSwNP, biomarker, gene expression, nasal polyp

## Abstract

TSLP is an alarmin released upon activation of epithelia in response to various external stimuli and is involved in type 2 cytokine-mediated pathological disorders. The formation of a high-affinity heterodimeric receptor complex, comprising the thymic stromal lymphopoietin receptor (TSLPR) chain and IL-7Rα, is required for signaling. This study investigated whether *TSLP* and *TSLPR* expression in peripheral blood or nasal polyps could provide a valuable approach for the molecular phenotyping of patients with chronic rhinosinusitis with nasal polyps (CRSwNP). The study population comprised 156 unrelated Caucasian individuals, including 45 controls and 111 patients with CRSwNP. Quantitative PCR analysis of *TSLP* and *TSLPR* was performed on the population study’s peripheral blood and nasal biopsy. The data were analyzed for potential associations, and possible use as a biomarker was studied. Significant differences were observed in *TSLP* and *TSLPR* blood expression between the control group and patients. Similarly, the expression of *TSLP* observed in biopsy samples was statistically significantly elevated in the polyp tissue of the patient compared with healthy controls. The combination of *TSLP* and *TSLPR* expression testing with peripheral blood eosinophils represents a more specific biomarker in patients exhibiting low eosinophil values. Further investigation of *TSLP*/*TSLPR* mRNA levels in peripheral blood may yield new minimally invasive biomarkers.

## 1. Introduction

Chronic rhinosinusitis (CRS) is a heterogeneous and persistent inflammatory condition affecting the nasal and paranasal sinuses. This disease is characterized by symptoms including nasal obstruction or congestion, nasal secretions, facial pain or pressure, and olfactory alterations [1]. CRS presents at least two main phenotypes: CRS with nasal polyps (CRSwNP) and CRS without nasal polyps (CRSsNP) [2]. CRS has an important and social burden with notable effect on quality of life, both in children and adults [2,3]. In children, hereditary syndromes such as cystic fibrosis or primary ciliary dyskinesia should be considered [2]. The degree of disease severity and its impact on a patient’s quality of life and social cost are determined by genetic, environmental, and behavioral factors and the presence of comorbidities [2].

CRSwNP has been categorized as type 2 (T2) and non-T2. T2 CRSwNP is more prevalent in Western countries and is associated with a worse prognosis; high eosinophil counts in the polyp tissue and blood, and high levels of T2 cytokines and chemokines [4]. Common T2 CRSwNP comorbidities are asthma and nonsteroidal anti-inflammatory drug (NSAID)-exacerbated respiratory disease (N-ERD) [3] under the common umbrella of a type 2 inflammatory pathway [5,6]. The inflammatory mechanism of CRSwNP is initiated by the activation of epithelial cells by different triggers, which subsequently induce the production of alarmins [7], which in turn stimulate the production of cytokines with a role in T2 inflammation [8].

Thymic stromal lymphopoietin (TSLP) is an alarmin released upon activation of the epithelia in response to various external stimuli. Two isoforms have been identified in humans: the long-form TSLP (lfTSLP) and the short-form (sfTSLP) [9]. Despite the distinct differences between the two isoforms of TSLP in humans, the specific functions of sfTSLP remain unclear [10]. lfTSLP has been involved in immune responses, especially in inflammation and allergy, while sfTSLP may play a homeostatic role, acting as an inflammation suppressor [11]. TSLP is produced by various cell types, including airway epithelial cells, dendritic cells, T cells, NK-T cells, eosinophils, mast cells [12,13,14], monocytes, macrophages, and granulocytes [10]. TSLP expression is induced in response to pathogenic stimuli upon contact with airway epithelial cells, including aeroallergens such as fungi, dust mites, cockroaches, and pollen [15].

TSLP signaling requires the formation of a high-affinity heterodimeric receptor complex comprising the thymic stromal lymphopoietin receptor (TSLPR) chain and IL-7Rα [14]. TSLP can exert biological functions by acting on a wide range of immune cells that express TSLP receptors [15]. These include dendritic cells (DCs), basophils, CD4 + T cells, and group 2 innate lymphoid cells (ILC2s) [16,17,18,19,20,21,22,23,24], which drive T2 inflammatory responses, such as asthma [25] or CRSwNP [26].

Upon binding its receptor complex, TSLP activates Janus kinases (JAK), which in turn activate the phosphorylation and signal transducer and activator of transcription (STAT), thereby initiating pro-inflammatory signaling [27]. In patients diagnosed with asthma and nasal polyps (NPs), TSLP activates fibroblasts and smooth muscle cells in conjunction with other cytokines, such as IL-1, IL-6, and TNF-alpha, to induce smooth muscle hypertrophy and increased remodeling of the bronchial and nasal epithelium [27]. Studies have demonstrated that *TSLP* mRNA expression is elevated in nasal polyps [28] and epithelial cells from individuals with asthma [29], atopic dermatitis [30], COPD (chronic obstructive pulmonary disease). Elevated expression also occurs in patients with rhinosinusitis, suggesting its potential involvement in the pathogenesis of CRSwNP [31,32]. Moreover, *TSLP* expression was elevated in the smooth muscle of the airway and lungs in patients with mild-to-moderate and severe asthma [33,34]. Additionally, increased levels of *TSLPR* have been detected in asthmatic children, in many patients with autoimmune and cancerous disorders, and in the eosinophils in the mucosa in patients with eosinophilic chronic rhinosinusitis (CRS) [35].

TSLP, IL-25, and IL-33 are key players in type 2 inflammatory responses associated with allergic rhinitis (AR), CRS, and asthma [36,37]. There is growing evidence that several single-nucleotide polymorphisms (SNPs) in the genes encoding these cytokines are associated with the development of asthma. This association has been demonstrated for SNPs located in the *TSLP* gene’s promoter region and SNPs in the *IL-33* gene [38,39,40,41]. Furthermore, SNPs in *TSLP* and *IL-33* have been shown to increase susceptibility to developing CRSwNP [42,43,44].

In addition, TSLP has been linked to other conditions, including atopic dermatitis and eosinophilic esophagitis (EoE) [45]. In this sense, a study demonstrated that specific SNPs and alleles of *TSLP* were associated with an increased risk of developing EoE [46]. However, TSLP levels in serum were not elevated in EoE patients compared to healthy controls [47]. Conversely, children and adults with atopic dermatitis exhibited significantly elevated serum TSLP levels compared to healthy individuals [48].

Moreover, epigenetic mechanisms, such as DNA methylation, have been identified as contributing factors in developing and worsening atopic diseases, including dermatitis, asthma, and allergic rhinoconjunctivitis [49]. Previous studies have demonstrated a negative correlation between *TSLP* methylation levels and *TSLP* expression [41,50,51,52,53,54]. Additionally, a preliminary study demonstrated that DNA methylation at the *TSLP* locus was associated with CRSwNP pathogenesis [55].

Considering the role of TSLP as a pivotal initiator of epithelial inflammatory responses to damage, research was driven to identify drugs that target the TSLP-TSLPR pathway [48,56]. The first investigational anti-TSLP medicine, tezepelumab, a human immunoglobulin G2λ monoclonal antibody that inhibits the interaction of TSLP with its heterodimeric receptor, has recently been approved [57]. Tezepelumab has been demonstrated to reduce exacerbation rates, improve lung function, and reduce multiple biomarkers of inflammation [48]. In the Phase III NAVIGATOR trial, subjects with severe asthma and CRSwNP who received tezepelumab significantly improved their Sinonasal Outcome Test-22 (SNOT-22) scores over a 52-week treatment period [58]. A multicenter Phase Ib/IIa clinical study for subjects with uncontrolled CRSwNP (DUBHE) was conducted to assess the safety, tolerability, pharmacokinetics, pharmacodynamics, immunogenicity, and efficacy of multiple ascending doses of another anti-TSLP monoclonal antibody, CM326 [59].

Regarding TSLPR, two Phase 2 clinical trials have been initiated with the monoclonal antibody anti-TSLPR named Verekitug (UPB-101). One trial has been undertaken in severe asthma patients and another in patients with chronic rhinosinusitis with nasal polyps. Verekitug is a novel recombinant fully human immunoglobulin G1 monoclonal antibody that blocks TSLPR and inhibits TSLP-driven inflammation. The Phase 1b trial in asthma patients demonstrated that Verekitug significantly affected exhaled nitric oxide and blood eosinophils [60,61,62].

In summary, considerable evidence suggests that TSLP is implicated in the pathogenesis of allergic and asthmatic diseases [13,63]. This evidence further indicates that *TSLP* levels might serve as a potential biomarker for phenotypic characterization, prognosis, and monitoring of treatment response [64]. Accordingly, this study aimed to investigate the involvement of *TSLP* and *TSLPR* expression in CRSwNP, CRSwNP with asthma, and N-ERD. To this end, the expression of *TSLP* and *TSLPR* mRNA in the blood and tissue of patients was compared to that of healthy individuals.

## 2. Results

### 2.1. Characteristics of the Population Study

The study population comprised 156 individuals; 45 were included in the control group, 41 patients were diagnosed with asthma with CRSwNP, 21were patients with N-ERD, and 49 were diagnosed as patients with CRSwNP without asthma. The characteristics of the study population are shown in Table 1.

No differences in age distribution between patients and healthy controls (HCs) were observed. However, the sex distribution showed significant differences between the two groups. As expected, atopy distribution was different between HCs and patients but not between the various subgroups of patients. Furthermore, IgE was significantly higher in the global patient group compared to HCs (*p* < 0.001), as well as when compared to each patient subgroup (CRSwNP without asthma: *p* = 0.003, N-ERD: *p* < 0.001, and asthmatics with CRSwNP: *p* < 0.001). On the other hand, peripheral blood eosinophils (PBE) were significantly higher in patients than in HCs (*p* < 0.001), as well as in all patient subgroups (*p* < 0.001), but we did not observe significant differences between patient subgroups. No significant differences were observed between patients regarding their SNOT22 or FeNO levels (Table 1). Moreover, these results were not affected by atopy in any patient subgroup.

### 2.2. TSLPR and TSLP Expression in Peripheral Blood Samples

Significant differences were observed in *TSLPR* blood expression between the controls and patients (*p* < 0.001) as well as between the controls and all patient subgroups (*p* < 0.001 for all comparisons). Similarly, *TSLP* blood expression also exhibited significant differences when comparing the control group with patients (*p* < 0.001) and with all patient subgroups (CRSwNP without asthma: *p* < 0.001, N-ERD: *p* = 0.001, and asthmatics with CRSwNP: *p* = 0.041) (Table 2; Figure 1). In the N-ERD subgroup of patients, a statistically significant correlation was observed between *TSLPR* and *TSLP* expression levels (r = 0.561, *p* = 0.008). A positive correlation tendency was noted in the remaining subgroups, though without reaching statistical significance.

### 2.3. Characteristics of the Population of the Biopsy Study

In order to ascertain the differences in expression between peripheral blood and nasal biopsy, 33 patients from the previous population study were analyzed. The biopsy cohort comprised 11 individuals with CRSwNP and asthma, 11 with N-ERD, 11 with CRSwNP without asthma, and 11 healthy controls. The characteristics of this population are described in Table 3.

Regarding the demographic and clinical characteristics of the subjects, no significant differences were observed in the distributions by sex and age between the control and patient subgroups. As expected, significant differences in atopy and IgE levels were found when comparing the HCs and the patient subgroups (*p* = 0.002). In addition, peripheral blood eosinophils (PBE) were significantly higher in the global patient cohort compared to HCs (*p* < 0.001) and in the subgroup of asthmatics with CRSwNP (*p* = 0.001) and N-ERD (*p* = 0.005) compared to the HCs. Nevertheless, no differences were observed between the HCs and the CRSwNP subgroup without asthma (*p* = 0.239). No significant differences were observed between the global patient group, patient subgroups, and the HCs for the remaining variables (Table 3). Furthermore, these results were not affected by atopy in any patient subgroups.

### 2.4. TSLPR and TSLP Expression in Nasal Biopsy Samples

The expression of *TSLPR* was observed to be elevated in the polyps of all patient groups compared to the nasal tissue of control subjects (7.01 ± 17.99 vs. 2.06 ± 1.44; *p* = 0.144) (Table 4). However, this increase was not found to be statistically significant. Notably, the highest expression was observed in the polyp tissue of patients with CRSwNP and asthma (13.98 ± 30.65; *p* = 0.341) (Table 4). This lack of statistical significance may be attributed to the relatively small sample size.

The expression of *TSLP* observed in biopsy samples was found to be significantly elevated in the polyp tissue of all patient subgroups in comparison with HCs (CRSwNP without asthma) (72.41 ± 53.64, *p* = 0.048); asthmatics with CRSwNP (104.49 ± 76.78, *p* = 0.009); and N-ERD (121.23 ± 145.03, *p* = 0.009), (Table 4). However, no statistically significant differences regarding *TSLP* were identified between groups in peripheral blood samples (Table 4 and Figure 2).

Furthermore, a correlation was observed between *TSLPR* and *TSLP* expression levels in both biopsy tissue (r = 0.377, *p* = 0.031) and blood samples (r = 0.442, *p* = 0.010).

When the subgroups of patients were analyzed, a correlation between *TSLPR* and *TSLP* expression in polyp samples was observed in the CRSwNP (r = 0.664; *p* = 0.026) and CRSwNP with asthma subgroups (r = 0.692; *p* = 0.018).

In the N-ERD subgroup of patients, a statistically significant correlation was observed between *TSLPR* and *TSLP* expression levels in blood samples (r = 0.845, *p* = 0.001). Additionally, a correlation between FeNO and SNOT 22 (r = 0.900; r = 0.037) was observed in this subgroup.

## 3. Discussion

This study investigated whether *TSLP* and *TSLPR* mRNA expression differed between patients and controls in peripheral blood and biopsy samples and whether there was a correlation between tissue and peripheral blood. Our results demonstrated that the expression of *TSLP* and *TSLPR* in peripheral blood samples was elevated in all patient subgroups compared to the control group, and that this increase was not influenced by atopy. These data support the hypothesis that an elevated expression of *TSLPR* and *TSLP* in the peripheral blood of patients could be associated with the inflammatory process that these patients are experiencing due to their diseases [4].

TSLPR exhibits a high binding affinity for TSLP but not for IL-7Rα, which requires the formation of the binary complex (TSLPR-IL-7Rα) to initiate intracellular signaling [65,66]. *TSLPR* is expressed in immune cells such as mast cells, NKT cells, and eosinophils, making them responsive to *TSLP* [67]. Moreover, TSLPR is upregulated or constitutively expressed in many patients with rheumatoid arthritis and acute lymphoblastic leukemia [35]. That has prompted research into TSLP, TSLPR, and its associated signaling pathways.

Many studies have focused on the expression of *TSLP* in several tissues, including the gut, skin, and lung [68]. Levels of *TSLP* in the tissue of asthmatic patients or keratinocytes of acute and chronic lesions of atopic dermatitis patients have been linked to disease severity [30,33]. The present study revealed a significant difference in *TSLP* expression levels between nasal polyp tissue and control nasal mucosa across all of the patient subgroups, whereas no such difference was observed in *TSLPR* expression. That could be because *TSLP* is expressed mainly in the epithelial cells of nasal mucosa [28]. Conversely, *TSLPR* is expressed mainly in immune cells such as dendritic cells, CD4+ cells, and ILC2s [15]. These findings indicate that *TSLP* and *TSLPR* expressions may be regulated differently in nasal polyps, suggesting that polyps may play a pivotal role as *TSLP* sources, enhancing the signaling through airway cells. Indeed, Kaur et al. showed that *TSLP* expression was elevated in airway smooth muscle (ASM) in patients with mild-to-moderate disease and that it activates mast cells, which increase chemokine and cytokine production [34]. Redhu et al. suggested that TSLP/TSLPR-mediated autocrine activation of ASM may be a contributing factor [69]. Nagarkar et al. provided evidence indicating that, in addition to the observed increase in mRNA *TSLP* levels, there was a significant elevation of *TSLPR* in polyp tissue from patients with CRSwNP compared to control subjects [28].

Furthermore, Buchheit et al. demonstrated that *TSLP* mRNA was also similarly detected in N-ERD and CRSwNP patients in nasal polyps [70]. Another study indicated that *TSLPR* expression was similarly elevated in CRSwNP and CRSwtNP (CRS without NP) [71], whereas *TSLP* mRNA expression was notably higher in individuals with N-ERD than CRSwtNP [72]. Moreover, it has been demonstrated that the increased expression of *TSLP* in nasal epithelial cells of patients with allergic rhinitis can be associated with developing nasal polyps [32]. Additionally, a comprehensive transcriptome RNA sequencing of 42 polyps (CRSwNP-NP), 33 paired nonpolyp inferior turbinate tissue samples from patients with polyposis (CRSwNP-IT), and 28 inferior turbinate tissue samples from controls (CS-IT) revealed the presence of gene signatures associated with impaired host defense, inflammation, and aberrant extracellular matrix metabolism in CRSwNP [73]. The results demonstrated that *TSLP* was differentially expressed in the comparison between CRSwNP-NP vs. CS-IT but not in CRSwNP-NP vs. CRSwNP-IT [72], which supports the findings of our study.

Interestingly, a correlation between *TSLP* and *TSLPR* levels was observed in tissue and blood samples. This finding suggests that measuring these gene expressions in blood could serve as a reliable and minimally invasive biomarker, which is desirable for a promising biomarker.

PBE counts have been suggested to be a biomarker for monitoring polyp growth in CRSwNP patients with eosinophilia, asthma, and/or N-ERD [74]. TSLP has been associated with a T2-high inflammatory profile of CRSwNP [4]. It has been reported that the expression of *TSLP* and *TSLPR* was upregulated in eosinophilic but not non-eosinophilic nasal polyps and epithelial cells from CRSwNP patients compared to controls [75,76]. TSLP has been demonstrated to promote the production of eotaxin-1 by nasal epithelial cells from CRSwNP patients via the JAK-STAT3 pathway, thereby contributing to eosinophilia and inflammation [77].

As outlined above, TSLP is predominantly implicated in sustaining T2 inflammation. Nevertheless, evidence suggests that TSLP plays a role in non-eosinophilic CRSwNP [4].

*TSLP* expression is complex and can be increased by other indicators. Wang et al. observed that oncostatin M (OSM) increased *IL-4Ra* expression, which induced *TSLP* synthesis in nasal epithelial cells (NECs) [78]. Similarly, Ogasawara et al. showed that TSLP enhanced RANK-L-mediated type 2 cytokine production from ILC2s in CRSwNP [79]. Liao et al. also observed that TSLP could induce interleukin-1 receptor-like 1 expression, which promoted IL-33-induced TSLP expression in human nasal epithelial cells (HNECs) [76]. Pathinayake et al. showed that endoplasmatic reticulum (ER) stress can increase TSLP production in bronchial biopsies, which is subsequently enhanced by TLR3 activation, a process that may contribute to severe asthma exacerbations [80]. In addition, studies in patients with CRSwNP have shown that increased DNA methylation at the *TSLP* locus is associated with the pathogenesis of CRSwNP [54]. Concerning protein expression levels, Poposki et al. demonstrated an association between a TSLP metabolite generated after carboxypeptidase N digestion and strong activation of group 2 myeloid dendritic cells and innate lymphoid cells compared to mature TSLP [81].

Recently, several humanized monoclonal antibodies have been developed for targeted therapies, including anti-TSLP (tezepelumab) [25,48,57,59] and anti-TSLPR (verekitug) as part of the immune system response pathways associated with chronic inflammation such as CRSwNP [60,61,62].

Tezepelumab effectively enhanced clinical outcomes in patients exhibiting both T2-high and T2-low phenotypes. The PATHWAY and NAVIGATOR studies have indicated that the subjects with eosinophilic inflammation displayed the highest reduction in the frequency of asthma exacerbation [82,83]. Treatment with tezepelumab significantly improved both asthma outcomes and sinonasal symptoms in patients with severe asthma and comorbid CRSwNP [58].

The determination of *TSLP* and *TSLPR* levels could be associated with responses to anti-TSLP or anti-TSLPR monoclonal antibodies. This hypothesis is attractive, although it is unexplored. The correlation between polyps’ tissue and peripheral blood level could be valuable in this sense. In addition, tezepelumab has also demonstrated efficacy in patients with low T2 biomarker levels [4], which could support the hypothesis of the utilization of TSLP/TSLPR expression. In this sense, biomarker studies such as the present work could provide valuable insights into applying personalized precision medicine through better patient classification to predict their response to these new treatments.

This study is not without limitations. One limitation is its unicentric nature. However, this feature ensured uniformity across the study and facilitated a comprehensive population characterization. The limited sample size may have constrained the sub-analyses capacity to detect significant differences between the control and patient groups. Consequently, these findings need to be confirmed in larger cohorts.

## 4. Materials and Methods

### 4.1. Study Population

All patients were recruited from the Allergy and Otorhinolaryngology Departments of the University Hospital of Salamanca. The study was approved by the Clinical Research Ethics Committee of the Institute for Biomedical Research of Salamanca (IBSAL) (PI 2020-02-433). The study was conducted following the recommendations of the Ethics Committee of the University Hospital of Salamanca. All participants signed a written informed consent form. Controls fulfilled the following criteria: (i) Absence of symptoms or history of asthma, nasal polyposis, N-ERD, or other pulmonary diseases, (ii) Absence of symptoms or history of rhinitis, (iii) Absence of symptoms or history of allergic diseases, (iv) Negative results on skin prick tests to a battery of common aeroallergens; (v) Absence of a family history of asthma, rhinitis, or atopy, and (vi) Age > 16 years old.

In addition, the patients were selected according to the following criteria: (i) A physician’s diagnosis of asthma, nasal polyposis, or N-ERD, and (ii) an age of greater than 16 years. CRSwNP and N-ERD were diagnosed according to EPOS criteria [1]. Asthma was diagnosed according to the GINA guidelines [84]. The severity of asthma was evaluated according to the Spanish Guide for the Management of Asthma (GEMA 5.4) [85]. Skin prick tests were conducted using a battery of common aeroallergens [86] based on recommendations from the European Academy of Allergy and Clinical Immunology (EAACI) [87]. Histamine 10 mg/mL was employed as the positive control, while saline 0.9% was the negative control. A positive result was a wheal at least 3 mm larger in diameter than the negative control. Patients were considered atopic if they exhibited a positive skin reaction to at least one allergen. None of the subjects were receiving oral corticosteroids.

Fractional exhaled nitric oxide (FeNO), total IgE, and lung function parameters were evaluated for all the patients. CRSwNP improvement was assessed utilizing the Sino-nasal Outcome Test (SNOT-22) [88].

Nasal polyps and mucosa tissue were biopsied from patients and healthy controls. Following the biopsy, tissues were immediately immersed in RNA Later and stored at −80 °C for later use.

### 4.2. Clinical Measurements

Peripheral blood eosinophils, basophils, leucocytes, monocytes, lymphocytes, and platelets were counted automatically using a counter (Beckman Coulter, Brea, CA, USA) and the MAXM A/L system (Beckman Coulter). Serum levels of total IgE were quantified by a fluoroenzyme immunoassay (ImmunoCap System, ThermoFisher Scientific, Waltham, MA, USA). The fractional exhaled nitric oxide (FeNO) was determined using NIOX VERO (Circassia, Uppsala, Sweden). In nasal polyps, hematoxylin/eosin staining was performed for eosinophil count with high field magnification (40×).

### 4.3. RNA Isolation and RT-PCR

Total RNA was isolated from peripheral blood samples stored with RNA Later at −20 °C, using the RiboPure-Blood kit (Ambion, Thermo Fisher Scientific, Waltham, MA, USA) according to the manufacturer’s instructions. DNAse treatment was conducted using Turbo DNAse (Ambion, Thermo Fisher Scientific, Waltham, MA, USA). The concentrations and RNA quality ratios were determined using a NanoDrop 1000 (Thermo Fisher Scientific, Waltham, MA, USA). Reverse transcription (RT) was performed on 500 ng of total RNA using the Superscript III First-Strand Synthesis System for RT-PCR (Invitrogen, Thermo Fisher Scientific, Waltham, MA, USA). The thermal cycler (MultiGene Opti-Max, Labnet International Inc., Edison, NJ, USA) was employed with a total volume of 20 µL, comprising a single cycle and incubation periods of 65 °C for 5 min, 25 °C for 10 min, 50 °C for 50 min, 85 °C for 5 min, and 37 °C for 20 min. All the samples were subjected to the same reverse transcription reaction conditions.

Furthermore, RNA extraction was conducted on polypoid tissue and healthy nasal mucosa in addition to the blood RNA isolation. A homogenizer (Fisherbrand™) was employed to disaggregate tissues from selected patients. The subsequent purification of the isolated RNA was performed using the RiboPure™ RNA Purification Kit (Ambion, Thermo Fisher Scientific, Waltham, MA, USA). The DNAse treatment was performed using Turbo DNAse (Ambion, Thermo Fisher Scientific, Waltham, MA, USA). The concentrations and RNA quality ratios were determined using a NanoDrop 1000 (Thermo Fisher Scientific, Waltham, MA, USA). RT was performed using the Superscript III First-Strand Synthesis System for RT-PCR (Invitrogen, Thermo Fisher Scientific, Waltham, MA, USA) with 1000 ng or 2000 ng of total RNA, as determined by the amount of starting genetic material. The thermal cycler (MultiGene Opti-Max, Labnet International Inc., Edison, NJ, USA) was employed for this process. The reaction mixture was prepared in a total volume of 20 µL, comprising a single cycle and incubation periods of 65 °C for 5 min, 25 °C for 10 min, 50 °C for 50 min, 85 °C for 5 min, and 37 °C for 20 min. All of the samples were subjected to the same reverse transcription reaction conditions.

### 4.4. Quantitative PCR Expression Analysis

Relative quantitative PCR (qPCR) was performed using the LightCycler480^®^ Instrument and SYBR Green I Master (Roche, Basel, Switzerland). The comparative Ct method was used to calculate the fold induction using the formula 2-(ΔΔCt) [89]. Primers for the *TSLP* receptor (*TSLPR*) and *TSLP* were designed using the software Primer 3.0 [90] and subsequently refined using the Beacon Designer software [91]. The *GAPDH* reference gene primers were selected from the Real-Time Ready Human Reference Gene Panel (Roche Applied Science, Indianapolis, IN, USA). The primer sequences used are presented in Table 5.

The efficacy of the primers was evaluated by amplifying serial dilutions of a cDNA sample with a known concentration in accordance with the following equation: E = (10−1/slope − 1) × 100. All reactions were performed in triplicate. The triplicates were considered valid if the standard deviation was less than 0.35. In each experiment, non-template controls and a calibrator were included. The PCR conditions included 10 min at 95 °C followed by 45 cycles of 10 s at 95 °C for denaturation, 10 s at 60 °C for annealing, and 10 s at 72 °C for polymerization. All procedures were conducted per the guidelines set forth by the Minimum Information for Publication of Quantitative Real-Time PCR Experiments (MIQE) [92].

### 4.5. Statistical Analysis

Data analysis and graphs were performed using the SPSS Software, version 28 (IBM, Armonk, NY, USA). The Kolmogorov–Smirnov Z test was employed to ascertain the normality of the distribution. The analysis of variance (ANOVA) with Dunnett’s multiple comparisons test (DMS) was employed to compare continuous parametric data, whereas the Kruskal–Wallis test was used for non-parametric data. In cases involving multiple comparisons, Bonferroni corrections were utilized to adjust the original *p*-values, considering the original *p*-value as *p* < 0.05 to calculate the new *p*-values. Proportions were compared using the Chi-squared test. Pearson and Spearman correlations were conducted to examine the relationships between variables.

The threshold for statistical significance was set at *p* < 0.05 for all analyses unless otherwise specified. Graphical representations of the data included box plots, which were used to facilitate visual interpretation and comparison between groups.

While the differences between the means and standard deviation of the groups could be large in some instances, analyses based on the difference between the distributions give a more reliable measure, since the data may contain outliers. For this purpose, different statistical tests that are more dependent or independent of the mean have been included.

The R library ‘pwr’ was used to calculate the statistical power with the formula: Power = 1 − β = P (t observed > t critical). β is the probability of making a type II error. The observed t is the value calculated from the data to measure the difference between groups, and the critical t is the threshold value of the t-distribution used to decide whether that difference was statistically significant at the chosen significant level.

## 5. Conclusions

In conclusion, our results indicate that *TSLP* and *TSLPR* play a role in inflammatory diseases such as CRSwNP and N-ERD. *TSLP* levels were significantly elevated in the peripheral blood and biopsies of patients compared to controls. However, our study showed a significant increase in *TSLPR* levels only in the peripheral blood of patients, possibly due to the small number of biopsy samples and the heterogeneity of the patients. Considering these findings, we propose that peripheral blood mRNA levels of *TSLP/TSLPR* should be investigated as potential new minimally invasive biomarkers that could assist in selecting patients for treatment with specific antagonists.

## Figures and Tables

**Figure 1 ijms-26-01227-f001:**
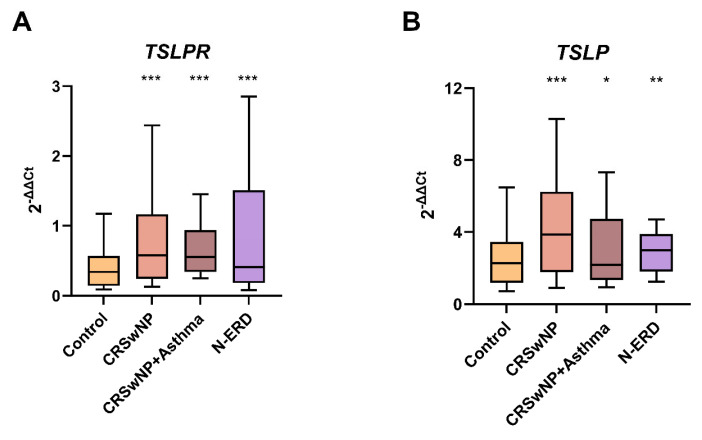
The boxplots illustrate (**A**) The expression of *TSLPR* in peripheral blood and (**B**) The expression of *TSLP* in peripheral blood. CRSwNP: chronic rhinosinusitis with nasal polyposis without asthma; N-ERD: NSAID-exacerbated respiratory disease; *** *p* < 0.001; ** *p* < 0.01; * *p* < 0.05; *p*-value of the Kruskal–Wallis test for each group of patients vs. controls.

**Figure 2 ijms-26-01227-f002:**
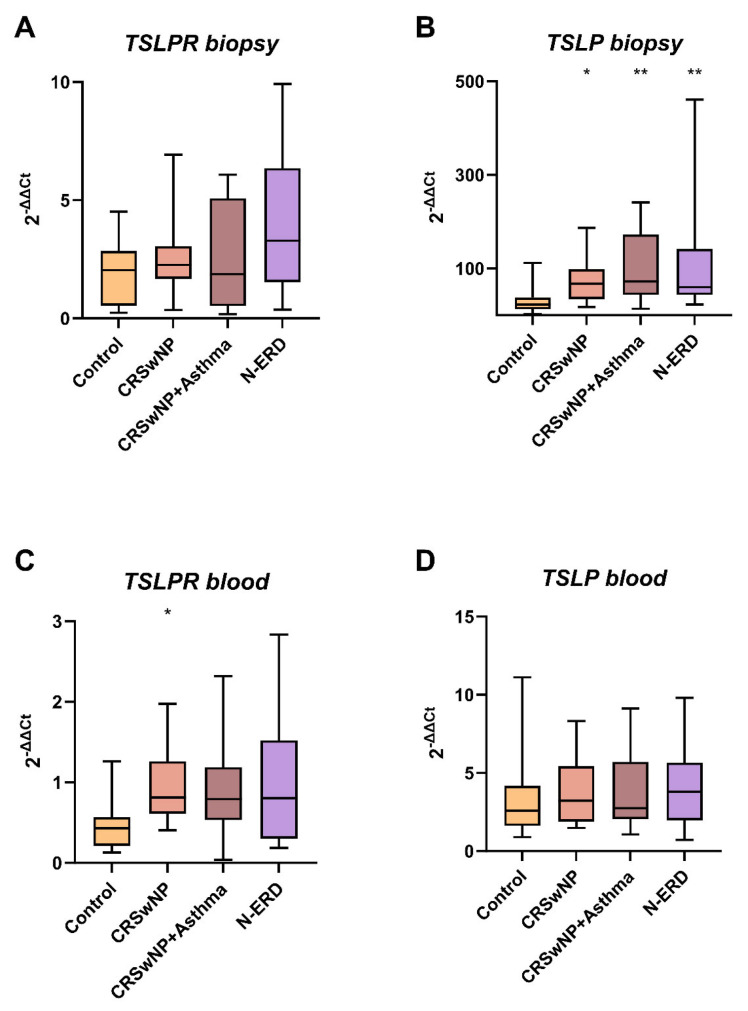
The boxplots illustrate (**A**) The expression of *TSLPR* in tissue biopsy, (**B**) The expression of *TSLP* in tissue biopsy, (**C**) The expression of *TSLPR* in peripheral blood, and (**D**) The expression of *TSLP* in peripheral blood. CRSwNP: chronic rhinosinusitis with nasal polyposis; N-ERD: NSAID-exacerbated respiratory disease; ** *p* < 0.01; * *p* < 0.05; *p*-value of the Kruskal–Wallis test for each patient’s group vs. controls.

**Table 1 ijms-26-01227-t001:** Clinical and phenotypic characteristics of the study population.

	HCs			PATIENTS	
		Total	CRSwNP	CRSwNP +Asthma	N-ERD
N	45	111	49	41	21
Age (y) (mean ± SD)	55.51 ± 18.76	54.68 ± 16.43	53.29 ± 17.41	55.98 ± 16.60	55.43 ± 14.04
Sex, F (%)	35 (77.8) ^a^	42 (37.8) *	10 (20.4) ^c^	20 (48.8) ^b^	12 (57.1) ^a,b^
Atopy (%)	0 ^a^	49 (44.1) *	18 (36.7) ^b^	21 (51.2) ^b^	10 (47.6) ^b^
Total IgE (kU/L)	41.97 ± 49.77	225.18 ± 424.56 *	141.09 ± 206.52 *	344.90 ± 628.37 *¥	179.15 ± 187.34 *
PBE(cells/µL)	133.02 ± 86.75	395.37 ± 362.82 *	327.28 ± 221.33 *	427.25 ± 433.64 *	483.81 ± 447.07 *
SNOT-22	-	48.93 ± 20.51	42.57 ± 17.72	53.91 ± 21.29	52.27 ± 22.62
FeNO (ppb)	-	67.89 ± 62.69	-	78.24 ± 67.03	49.45 ± 49.11

Values are expressed as the mean ± standard deviation or percentage (%). Data were analyzed using Kruskal–Wallis analysis and adjusted using Bonferroni correction. Only statistically significant differences are indicated: (*): *p* < 0.05 compared to HCs; and (¥): *p* < 0.05 compared to the CRSwNP group. The superscript (a,b,c) denoting each letter represents a subset of group categories whose column proportions do not differ significantly from each other at the 0.05 level. N: Number; HCs: healthy controls; CRSwNP: chronic rhinosinusitis with nasal polyposis without asthma; N-ERD: NSAID-exacerbated respiratory disease; PBE: peripheral blood eosinophils: FeNO: Fraction of exhaled nitric oxide; ppb: parts per billion; SNOT-22: Sinonasal outcome test.

**Table 2 ijms-26-01227-t002:** *TSLPR* and *TSLP* expression levels in peripheral blood samples of the study population.

	HCs	PATIENTS			
		Total	CRSwNP	CRSwNP +Asthma	N-ERD
N	45	111	49	41	21
*TSLPR*	0.45 ± 0.47	1.01 ± 0.97 *****	1.11 ± 1.11 *****	0.81 ± 0.57 *****	1.18 ± 1.18 *****
*TSLP*	2.15 ± 1.45	4.31 ± 3.86 *****	4.99 ± 4.68 *****	3.61 ± 2.90 *	4.09 ± 3.27 *

Values are expressed as the mean ± standard deviation. Gene expression levels were determined by qPCR (2^−ΔΔCt^), and the mean and standard deviation were presented. Data were analyzed using Kruskal–Wallis analysis and adjusted using Bonferroni correction. Only statistically significant differences are indicated: (*): *p* < 0.05 compared to healthy controls (HCs). N: Number; CRSwNP: chronic rhinosinusitis with nasal polyposis without asthma; N-ERD: NSAID-exacerbated respiratory disease; *TSLP*: Thymic stromal lymphopoietin; *TSLPR*: *TSLP* receptor. The statistical power for comparisons was 0.95.

**Table 3 ijms-26-01227-t003:** Clinical and phenotypic characteristics of the study population with nasal biopsy samples.

	HCs	PATIENTS			
		Total	CRSwNP	CRSwNP +Asthma	N-ERD
N	11	33	11	11	11
Age (y) (mean ± SD)	54.36 ± 17.05	54.55 ± 16.93	55.09 ± 18.67	55.00 ± 18.24	53.55 ± 15.29
Sex, F (%)	9 (81.8) ^a^	19 (57.6)	4 (36.4) ^a^	8 (72.7) ^a^	7 (63.6) ^a^
Atopy (%)	0 ^a^	20 (60.6) *****	6 (54.5) ^b^	7 (63.6) ^b^	7 (63.6) ^b^
Total IgE (kU/L)	39.52 ± 71.50	220.59 ± 276.05 *****	172.87 ± 307.36	280.70 ± 308.46 *****	212.96 ± 216.88 *****
PBE (cells/µL)	114.55 ± 97.20	345.94 ± 185.85 *****	250.91 ± 83.60	391.00 ± 150.51 *****	400.00 ± 254.01 *****
EO biopsy	-	71.84 ± 64.67	57.50 ± 49.67	99.14 ± 81.39	60.50 ± 60.46
SNOT-22	-	51.14 ± 22.70	46.50 ± 11.47	49.00 ± 28.58	56.50 ± 24.95
FeNO (ppb)	-	54.58 ± 48.35	-	67.83 ± 56.95	47.20 ± 39.87

Values are expressed as mean ± standard deviation or percentage (%). Data were analyzed using Kruskal–Wallis analysis and adjusted using Bonferroni correction. Only statistically significant differences are indicated: (*): *p* < 0.05 compared to healthy controls (HCs); Each letter in the superscript (a,b denotes a subset of group categories whose column proportions are not significantly different at the 0.05 level. N: Number; CRSwNP: chronic rhinosinusitis with nasal polyposis without asthma; N-ERD: NSAID-exacerbated respiratory disease; PBE: peripheral blood eosinophils: FeNO: Fraction of exhaled nitric oxide; ppb: parts per billion; SNOT-22: Sinonasal outcome test.

**Table 4 ijms-26-01227-t004:** *TSLPR* and *TSLP* relative expression in the study population with nasal biopsy samples.

	HCs	PATIENTS			
		Total	CRSwNP	CRSwNP +Asthma	N-ERD
N	11	33	11	11	11
*TSLPR* biopsy	2.06 ± 1.44	7.01 ± 17.99	2.79 ± 2.15	13.98 ± 30.65	4.25 ± 3.24
*TSLP* biopsy	34.14 ± 34.35	99.37 ± 98.68 *****	72.41 ± 53.64 *****	104.49 ± 76.78 *****	121.23 ± 145.03 *****
*TSLPR* blood	0.51 ± 0.38	0.97 ± 0.68 *****	1.00 ± 0.53 *****	0.91 ± 0.69	1.01 ± 0.86
*TSLP* blood	3.56 ± 3.30	3.95 ± 2.54	3.84 ± 2.26	3.86 ± 2.73	4.14 ± 2.84

Values are expressed as the mean ± standard deviation or percentage (%). Gene expression values were determined by qPCR (2^−ΔΔCt^); the mean and standard deviation are presented. Data were analyzed using Kruskal–Wallis analysis and adjusted using Bonferroni correction. Only statistically significant differences are indicated: (*): *p* < 0.05 compared to healthy controls (HCs). N: Number; CRSwNP: chronic rhinosinusitis with nasal polyposis without asthma; N-ERD: NSAID-exacerbated respiratory disease; *TSLP*: Thymic stromal lymphopoietin; *TSLPR*: *TSLP* receptor. The statistical power for comparisons was 0.75.

**Table 5 ijms-26-01227-t005:** Sequences of primers used in the qPCR assay.

Primer		Sequence 5′-3′
*TSLP*	Forward	CGTAAACTTTGCCGCCTATGA
	Reverse	TTCTTCATTGCCTGAGTAGCATTTAT
*TSLPR*	Forward	AAGCGACTGGTCAGAGGTGACA
	Reverse	GAGGAGAGACACCATCAGAAGG
*GAPDH*	Forward	CTCTGCTCCTCCTGTTCGAC
	Reverse	ACGACCAAATCCGTTGACTC

*GAPDH*: Glyceraldehyde 3-phosphate dehydrogenase; *TSLP*: Thymic stromal lymphopoietin; *TSLPR*: *TSLP* receptor.

## Data Availability

The datasets presented in this article are not readily available because the data are part of an ongoing study.

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
