# Peer review of "TSLP and TSLPR Expression Levels in Peripheral Blood as Potential Biomarkers in Patients with Chronic Rhinosinusitis with Nasal Polyps"

_ijms, 2025, doi:10.3390/ijms26031227_

Round 1

Reviewer 1 Report

Comments and Suggestions for Authors

This submission described the potential of TSLP and TSLPR expression levels in peripheral blood as the biomarkers in patients with chronic rhinosinusitis with nasal polyps. The unfolding of the experiments was in a rational order. I recommend a major revision.

1. The authors should consult the gold standard of sentencing the corresponding diseases, and compare the findings in this work with the established standard.

2. If possible, more similar references should be cited to support the feasibility of using TSLP and TSLPR expression levels as biomarkers.

3. As mRNA indicators, TSLP and TSLPR expression should be checked to confirm the stability.

4. In the discussion or the last part of the experiments, the authors should raise several other indicators to cooperate with TSLP and TSLPR expression.

5. The language use should be improved.

Comments on the Quality of English Language

Moderate revision needed.

Author Response

Reviewer 1:

This submission described the potential of TSLP and TSLPR expression levels in peripheral blood as biomarkers in patients with chronic rhinosinusitis with nasal polyps. The unfolding of the experiments was in a rational order. I recommend a major revision.

Comments 1: The authors should consult the gold standard of sentencing the corresponding diseases and compare the findings in this work with the established standard.

Response 1:  We are grateful for the feedback provided by the reviewer. The selection of patients was conducted within the Allergy and Otorhinolaryngology Service at the Hospital of Salamanca, following the criteria of Global Initiative for Asthma (GINA) and European Position Statement on Otolaryngology (EPOS) guidelines. Considering the recommendations, we have implemented changes, and the correspondent references have been incorporated into the Material and Methods section part 4.1. (Page 9, Lines 872-888).

Comments 2: If possible, more similar references should be cited to support the feasibility of using TSLP and TSLPR expression levels as biomarkers.

Response 2: We appreciate the reviewer’s comment. Following another reviewer’s recommendation, the part relative to the use as a biomarker has been deleted. We have included information related to TSLP mRNA expression levels in the Introduction section part 1. (Page 2, lines 82-90).

Comments 3: As mRNA indicators, TSLP and TSLPR expression should be checked to confirm the stability.

Response 3: We thank the reviewer for his comment. In our experiments, we used GAPDH as a reference gene, selected from the Real-Time Ready Human Reference Gene Panel (Roche Applied Science, Indianapolis, IN, USA), and included negative controls and a calibrator in each experiment to ensure the accuracy and reliability of the results. Additionally, we performed primer efficiency studies for TSLP and TSLPR genes. These studies confirmed that the primers used had adequate efficiency for the concentration of cDNA, ensuring that the amplification of TSLP and TSLPR mRNAs was performed accurately, and that the data obtained were robust. Normalization was performed using GAPDH as a reference gene, and the results were validated with appropriate controls to ensure the stability of the expression of these genes. This information was included in the manuscript in Materials and Methods section part 4.4 (page 10-11; lines 1101-1131; Table 5).

We hope this information adequately addresses the reviewer's concern about the stability of the mRNA indicators, and we appreciate their valuable suggestions.

Comments 4: In the discussion or the last part of the experiments, the authors should raise several other indicators to cooperate with TSLP and TSLPR expression.

Response 4:

We thank the reviewer for the valuable comment. We agree that it is important to highlight additional indicators that may cooperate with TSLP and TSLPR expression. In the revised manuscript, we have incorporated several studies that explore factors and molecular signals interacting with TSLP.

This information is included in the Discussion section part 3. of the manuscript (page 9, lines 837-850).

Comments 5: The language use should be improved.

Response 5: We are grateful for this commentary. The manuscript's language has been in-depth revised, and the necessary changes have been made to improve its readability.

Reviewer 2 Report

Comments and Suggestions for Authors

DearEditor

Thanks a lot for hard work. I read this article with interest. However, I have some concerns.

Kindly incorporate the responses within the manuscript to enhance its overall quality.

The differences between chronic sinusitis in adults and children and the impact of chronic sinusitis on quality of life in adults and children should be stated in the introduction.

In the results, very small groups n=0-54 are analyzed. a twofold increase in size is necessary to draw statistically significant conclusions in order to obtain adequate EBM power.

Literature should be supplemented with: https://doi.org/10.3390/jpm13040618; https://doi.org/10.3390/children8121133

Comments on the Quality of English Language

There are many spelling and language mistakes and the manuscript needs to be corrected by a native English speaker.

Author Response

Reviewer 2:

Thanks a lot for hard work. I read this article with interest. However, I have some concerns. Kindly incorporate the responses within the manuscript to enhance its overall quality.

Comment 1: The differences between chronic sinusitis in adults and children and the impact of chronic sinusitis on quality of life in adults and children should be stated in the introduction.

Response 1: We thank the reviewer for the valuable comment. We have included the suggested information and the corresponding references in the Introduction section, page 1-2, lines 43-50.

Comment 2: In the results, very small groups n=0-54 are analyzed. a twofold increase in size is necessary to draw statistically significant conclusions in order to obtain adequate EBM power.

Response 2: We appreciate the comment. While the number of samples in the study may appear low, several factors should be highlighted. Firstly, the study population was rigorously selected, with all patients having CRSwNP. This condition is characterized by a lower degree of variability in the statistical distributions of data compared to studies with more heterogeneous populations. Rather than the number of samples, the relevant indicator to be considered here is the power of the tests. The R library 'pwr' has been used to calculate the power of the tests under the formula:

Power = 1- β = P (t observed > t critical). β is the probability of making a type II error (not rejecting the null hypothesis when it is false). The observed t is the value calculated from the data to measure the difference between groups, and the critical t is the threshold value of the t-distribution used to decide whether that difference is statistically significant at the chosen significance level.

As a result, the power of the test in the blood and tissue sample cohorts was 0.95 and 0.75, respectively, which are considered standard accepted values or even higher.

We agree with the reviewer that including more samples in the study would be desirable and that our findings must be confirmed in larger cohorts. Nevertheless, given their power, our tests' accuracy is not undermined by the population size.

We also included the following formula and text in the Material and Method section:

The R library 'pwr' has been used to calculate the power and the formula:

Power = 1- β = P (t observed > t critical). β is the probability of making a type II error (not rejecting the null hypothesis when it is false). The observed t is the value calculated from the data to measure the difference between groups, and the critical t is the threshold value of the t-distribution used to decide whether that difference is statistically significant at the chosen significance level. Page 11, lines 1149-1153.

We also added the calculated statistical power values in Tables 2 and 4.

Comment 3: Literature should be supplemented with: https://doi.org/10.3390/jpm13040618; https://doi.org/10.3390/children8121133

Response 3: Both references have been added as suggested, page 1-2, lines 43-50.

Comment 4: Comments on the Quality of English Language: There are many spelling and language mistakes and the manuscript needs to be corrected by a native English speaker.

Response 3: We are grateful for this commentary. The manuscript's language has been in-depth revised, and the necessary changes have been made to improve its readability.

Reviewer 3 Report

Comments and Suggestions for Authors

Dear editor, dear authors

a study on TSLP and TSLPR in CRS patients remains of interest due to the antibodies that have been developed lately for these factors

I have several comments though

I think that several references and comments that are included in the discussion should have been reported in the introduction. there is of no point to hide the information on related trials in the introduction and to keep this information for the discussion. The discussion should focus mostly on commenting about their results for example why they think that the predictive value of their model is increased when these factors are combined with eosinophils or why concentrations of TSLP between the various disease categories varies a little bit (e.g those with asthma have lower concentrations ) 

One of their central conclusions that the above factors can be utilized for the exclusion of nasal polyposis is maybe of minimal (if any) clinical utility since this disease can be easily excluded by nasal endoscopy. In addition I do not think that their conclusion in lines 409, 410 that these factors can be utilized for more accurate patient classification and improved treatment strategies, is supported by their results. Why they report this?

To my opinion of greater interest is that blood concentrations of TSLP are in line with the biopsy concentrations.

Other minor comments

Are TSLP and TSLPR expression related to SNOT22 or other measures?

I have the feeling that N-ERD patients usually have worse symptoms (increased SNOT22) which is not the case here

maybe tables 5.6.9.10 and figures 3 and 4 should be included as supplements

Author Response

Reviewer 3:

A study on TSLP and TSLPR in CRS patients remains of interest due to the antibodies that have been developed lately for these factors

Comments 1: I think that several references and comments that are included in the discussion should have been reported in the introduction. There is of no point to hide the information on related trials in the introduction and to keep this information for the discussion. The discussion should focus mostly on commenting about their results for example why they think that the predictive value of their model is increased when these factors are combined with eosinophils or why concentrations of TSLP between the various disease categories varies a little bit (e.g those with asthma have lower concentrations ).

Response 1: We appreciate this comment. To address this issue, we have incorporated several references related to clinical trials in the Introduction section part 1 to clarify this issue (page 3, lines 320-332). Discussion section (page 9, lines 851-860).

The Discussion section part 3 (page 8, lines 713-721) has been expanded to include further information that supports the association of TSLP and TSLP with eosinophil inflammation T2.

Comments 2: One of their central conclusions that the above factors can be utilized for the exclusion of nasal polyposis is maybe of minimal (if any) clinical utility since this disease can be easily excluded by nasal endoscopy. In addition I do not think that their conclusion in lines 409, 410 that these factors can be utilized for more accurate patient classification and improved treatment strategies, is supported by their results. Why they report this?

Response 2: We appreciate your comment and agree with your assessment. It is acknowledged that conventional diagnostic methods, such as nasal endoscopy and in some cases computed tomography, are the gold standard for identifying nasal polyps. We have included the information regarding to EPOS criteria for diagnosing CRSwNP into the Material and Methods section part 4.1 (page 9, lines 872-885).

Although our initial intention was to explore the use of TSLP and TSLPR and eosinophils as putative biomarkers, following the reviewer’s second and third comments, we propose deleting all parts relative to the use of TSLP and TSLPR together with eosinophil levels as a biomarker into de Results section part 2.5 and 2.6 and focusing the manuscript on the correlation of blood and tissue concentration of TSLP.

Comments 3: To my opinion of greater interest is that blood concentrations of TSLP are in line with the biopsy concentrations.

Response 3: We are grateful for the reviewer's observation. We totally agree with this comment. (see answer to comment #2). This point is emphasized in the Discussion section part 3. (page 8, lines 708-711).

Comments 4: Are TSLP and TSLPR expression related to SNOT22 or other measures?

Response 4: We thank the reviewer for his question. We have analyzed the correlation between TSLP/TSLPR and SNOT22, in addition to other T2 inflammatory biomarkers, such as peripheral blood eosinophils count, FeNO, IgE and these did not reach statistical significance.

This analysis revealed a statistically significant correlation between TSLP and TSLPR levels across the tissue and blood sample cohorts.

These observations are addressed in the Results section 2.2 (page: 4, lines 498-501), section 2.4 (pages: 6, lines: 591-594) and Discussion section 3 (pages: 8, lines: 713-721).

Comments 5: I have the feeling that N-ERD patients usually have worse symptoms (increased SNOT22) which is not the case here.

Response 5: Thank you for pointing this out. After reviewing the data, we identified an error in the table layout that could have led to a misinterpretation of the symptoms of patients with N-ERD.

We have corrected this error in Results section 2.1 (pages: 3,4; Table 1), and the data now accurately reflect the severity of symptoms in patients with N-ERD. According to the new analyses, we have observed that, as expected, patients with N-ERD tend to have more severe symptoms and higher SNOT22 scores than the other groups.

We thank the reviewer for pointing this out and trust that the corrections will remove any confusion.

Comments 6: maybe tables 5.6.9.10 and figures 3 and 4 should be included as supplements.

Response 6: We thank the reviewer for this suggestion. In agreement with the previous suggestion of deleting all data related to the putative use of TSLP and TSLPR as biomarkers, all these figures and tables have been deleted.

Round 2

Reviewer 1 Report

Comments and Suggestions for Authors

The authors have improved the submission a lot. I recommend the acceptance.

Author Response

Reviewer 1:

The authors have improved the submission a lot. I recommend the acceptance.

Author’s Response: We are grateful to the reviewer for the suggestions, which have significantly improved the manuscript.

Reviewer 2 Report

Comments and Suggestions for Authors

Dear Authors 

Thank you for correcting the work.

Please remember that statistics are supposed to describe reality, not create it.                      It seems to me that the authors here have succumbed to the temptation to create reality.  

I would ask the authors to familiarise themselves with evidence-based medicine(EBM). 

I suggest that they include a section in the paper about the limitations of the work and mention, among other things, the small number of cases studied.

Author Response

Author’s Response: We appreciate your commentaries improving the manuscript.

Please remember that statistics are supposed to describe reality, not create it.                      It seems to me that the authors here have succumbed to the temptation to create reality.

We cannot agree more that statistics are meant to describe reality and not to create it. Respectfully, the statistical tests have been adequate for this type of study, including, as suggested, statistical power. Although the first analysis of the group samples yielded a power of 0.95, which is within expectations, the second result of 0.75, although close to the recommended threshold of 0.8, highlights room for improvement and reinforces the importance of increasing the sample size. All analyses have been performed and supervised by an expert in biostatistics and bioinformatics. 

I would ask the authors to familiarise themselves with evidence-based medicine (EBM).

Concerning EBM, we are aware of the studies' rank. Notwithstanding, case-control studies are one of the major observational study designs for performing clinical research, which can be useful for studying disease outbreaks, rare diseases, or outcomes of interest (Dey T, Mukherjee A, Chakraborty S. A Practical Overview of Case-Control Studies in Clinical Practice. Chest. 2020 Jul;158(1S):S57-S64. doi: 10.1016/j.chest.2020.03.009. PMID: 32658653).

I suggest that they include a section in the paper about the limitations of the work and mention, among other things, the small number of cases studied.

We appreciate your suggestion of including limitations of our work, specifically mentioning the small sample and the need for larger cohorts to replicate the findings. We included the following paragraph on page 9, lines 343-347:

“This study is not without limitations. One limitation is its unicentric nature. However, this feature ensured uniformity across the study and facilitated a comprehensive population characterization. The limited sample size may have also constrained the sub-analyses capacity to detect significant differences between the control and patient groups. Consequently, these findings need to be confirmed in larger cohorts.”

Reviewer 3 Report

Comments and Suggestions for Authors

I believe that authors answered all my comments and their manuscript bacame more friendly for a rhinologist. The issue of TSLP and TSLPR expression in chronic rhinosinusitis is still under investigation and in this respect their data maybe of use and of interest for the readers

Author Response

Reviewer 3:

I believe that authors answered all my comments, and their manuscript became more friendly for a rhinologist. The issue of TSLP and TSLPR expression in chronic rhinosinusitis is still under investigation and in this respect their data may be of use and of interest for the readers

Author’s Response: We are grateful to the reviewer for the suggestions which have significantly improved the manuscript.
